# Corrosion Behavior of ZnMn Coatings Magnetoelectrodeposited

**Lamia Allam [1,2,*], Florica S. Lazar [1] and Jean-Paul Chopart [1]**

[1] Matériaux et Ingénierie Mécanique (MATIM), Université de Reims Champagne-Ardenne, BP 1039, CEDEX 02, 51687 Reims, France; florica.lazar@univ-reims.fr (F.S.L.); jean-paul.chopart@univ-reims.fr (J.-P.C.)

[2] Laboratoire de Physique et Chimie des Matériaux (LPCM), Université Mouloud Mammeri de Tizi-Ouzou, BP 17, Tizi-Ouzou 15000, RP, Algeria

* Correspondence: allamlamia@yahoo.fr

**Abstract:** The zinc–manganese alloy coatings have been obtained without and with superimposition of a 0.3 T magnetic field in a parallel direction to the working surface electrode. The electrodeposition during 30 min, for two applied potentials (E = −1.6 V/SCE and E = −1.8 V/SCE) in an electrochemical bath with the $(Zn^{2+})/(Mn^{2+})$ concentration ratio equal to 0.5. The structural, the morphological, and the chemical composition characteristics of the deposits have been studied. It has been found that the applied potentials modify the structural properties of the deposits, η phase-rich deposits elaborated for E = −1.6 V/SCE, and $MnZn_3$-rich deposits elaborated for E = −1.8 V/SCE. The magnetohydrodynamic convection favors the manganese content of the deposit. The corrosion behavior of these coatings has been analyzed in 3.5% NaCl solution by free corrosion potential measurements and electrochemical impedance spectroscopy. The different results show that the corrosion resistance of these zinc–manganese alloy coatings is linked to their structure, to their composition, and to the magnetic field amplitude used during the electrodeposition process.

**Keywords:** ZnMn alloys; corrosion; magnetic field; $MnZn_3$; η phase; EIS; XRD

## 1. Introduction

Since the replacement of cadmium coatings that are highly toxic [1], solutions have been oriented towards alloy zinc plating as a good protector of steel [2]. Zinc-based alloys have long been used as an anti-corrosion coating for products of different shapes: sheets, strips, tubes, screws, cables, and so on [3]. Zinc–manganese alloy coatings are anodic coatings, for which the free potential is more negative than that of the part to be protected (steel). Several studies have reported that, in an aggressive environment where sodium chloride and sulfur dioxide are present [4,5], the zinc–manganese alloy coating may show good corrosion resistance [6–8]. According to the literature, several properties, namely, composition, morphology, and structural properties, of deposits strongly affect their corrosion behavior. It has been reported by Gabe [3], Ortiz et al. [9], and Tomić et al. [10] that deposits with a Mn content ranging from 10% to 30% show good corrosion resistance.

However, Sylla et al. [11] obtained dendritic and growing powdery coatings rich in Mn that cannot be used as anti-corrosion coatings. The high Mn content in the Zn–Mn deposit is not a sufficient condition to improve corrosion resistance [12–14]. A combination of a high Mn content and a good morphology is necessary. The deposit morphology has a decisive effect on corrosion resistance, compared to the chemical composition [10,13–16].

The structure of the deposit also affects the corrosion behavior; referring to the literature, it seems that ε-phase ($MnZn_3$) monophasic deposits have higher stability against corrosion [7,11,12,17,18]. Indeed, the presence of a single phase in the coating reduces the risk of pitting or selective corrosion and the presence of more than two phases induces different corrosion behaviors and selective corrosion takes place for the least noble metal; the author adds that the interest of manganese in the alloy coating is related to the formation of particularly stable $Mn_2O_3$ oxide, which is detectable during natural corrosion

in the marine medium [19]. Several works [8,14,20] confirm that $Mn_2O_3$ oxide inhibits the cathodic oxygen reduction reaction and, thus, decreases the corrosion phenomenon. However, Bučko et al. [12] showed that the corrosion stability of alloys was increased when the hydroxy zinc chloride ($Zn_5(OH)_8Cl_2 \cdot H_2O$) was the majority product formed during the corrosion of Zn–Mn alloys. This compound has been shown to form better with zinc–manganese alloys than pure zinc [21–23]. In addition, the surface irregularity of the deposits obtained at high current densities has a negative effect on the stability of the passive film of zinc hydroxy chloride on the surface of the coating compared to those obtained at low current density. Considering the high number of reports in the literature on the electrodeposition of Zn–Mn alloy coatings, there has been little investigation into the corrosion behavior of these coatings, and no previous corrosion study has been reported for these alloys elaborated under magnetic field. In another paper of zinc–nickel alloys electrodeposited under magnetic field, Chouchane et al. [24] studied the electrochemical behavior in a 3% sodium chloride medium of these coatings. The authors found that when the magnetic field is superimposed at 12 T during electroplating, the morphology of the deposit is not largely modified, but the hydrogen reduction current on the deposits is decreased and, consequently, the corrosion potential of these alloys is modified. This phenomenon is concurrent with a decrease in the corrosion current. Akshatha R. Shetty et al. [25] founded that parallel and perpendicular magnetic field reduced the corrosion rate of Ni–Mo–Cd coatings. The effect is more pronounced in the case of the perpendicular, due to the effect of the Lorentz force.

The goal of the present paper is to compare the corrosion behavior of zinc–manganese alloy coatings elaborated without and with magnetic field for B = 0.3 T in NaCl 3.5 % and to relate the morphology, chemical, and phase composition of the deposits to their corrosion resistance.

## 2. Materials and Methods

### 2.1. Coating Elaboration

The working electrode was a disk of mild steel substrate used in our first paper [26]. Before each electrochemical deposition, the substrate surface was mechanically polished under water jet with silicon carbide paper (P120, 320, 600, and 1200) and, finally, rinsed with distilled water. To prepare the coatings, all the experimental conditions were the same of those mentioned in our first paper [26], namely the bath ratio chosen is $(Zn^{2+})/(Mn^{2+})$ = 0.5, at two potentials E = −1.6 V/SCE and E = −1.8 V/SCE with two magnetic field amplitudes B = 0 T and B = 0.3 T, with the exception of the deposition time, which was fixed at 30 min. The coatings were characterized by X-ray diffraction (XRD) (D8 Advance Bruker) equipped with Cu Kα radiation (2θ range of 30–90°, step time = 0.06°, and scan rate = 3 s by step) to determine phase composition. The chemical composition (atomic percentage at (%) in this paper) and morphological properties of the Zn–Mn coatings have been characterized by scanning electron microscopy (SEM) (JEOL JSM 6460LA microscope) coupled with the energy dispersive X-ray spectroscopy analysis (EDS) (JEL 1300 microprobe).

### 2.2. Corrosion Behavior

Corrosion measurements were carried out using a three-electrodes cell at room temperature and in an aerated medium NaCl at 3.5% without superimposed magnetic field. There was no stirring of the medium during these tests. This medium was appropriate for studying corrosion due to the presence of corrosion activators (chloride ions) [10]. The working electrode was the steel substrate recovered with the Zn–Mn coating of 0.95 $cm^2$ area, a platinum thick wire was used as the counter electrode (CE), and a saturated calomel electrode (SCE) was used as the reference electrode. The electrochemical measurement data were analyzed by Voltamaster 4 electrochemical software. The corrosion behavior of coatings was, firstly, studied by following open-circuit potential ($E_{ocp}$) during the immersion time until the potential of steel was reached. The electrochemical impedance spectroscopy (EIS) spectra were recorded after 2 h of immersion in corrosive medium, the

frequency range varied from 100 kHz to 10 mHz with 10 frequency points per decade, and the amplitude of the potential perturbation was equal to 10 mV. The numerical values of the parameters of the equivalent circuit were simulated using ZSimpWin software. The corrosion products powder was characterized by XRD.

## 3. Results and Discussions

### 3.1. Coating Elaboration

The study of the current density obtained without and with the magnetic field superimposition during electrodeposition for the applied potentials E = −1.6 V/SCE and E = −1.8 V/SCE revealed that whatever the magnetic field amplitude, there was a constant evolution of the current over time for the lower cathodic potential and a more stable current for the higher cathodic potential (Figure 1).

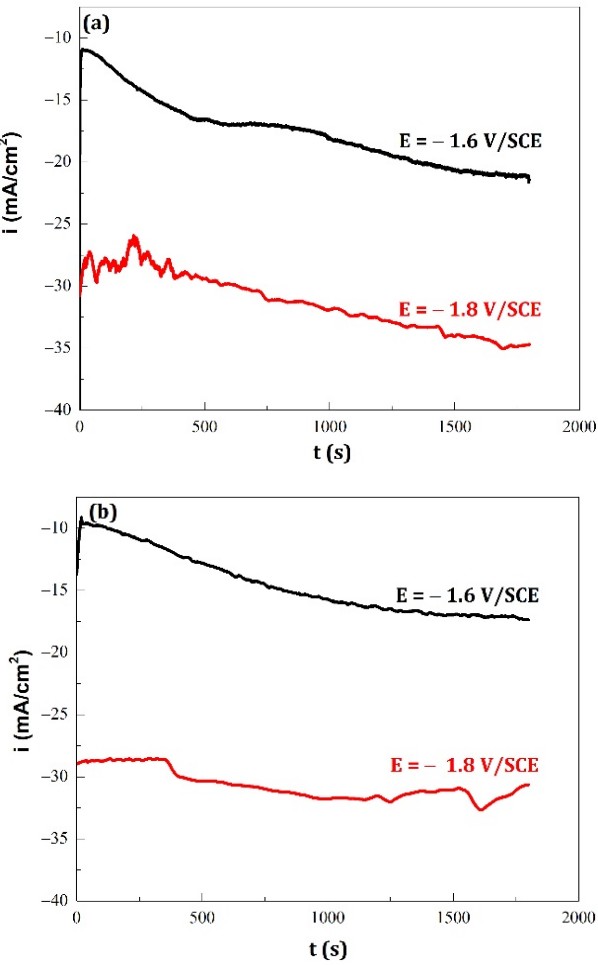

**Figure 1.** Current density (i) versus time (t) curves recorded during Zn–Mn alloy coatings electrodeposited as a function of magnetic field amplitudes: (**a**) B = 0 T, (**b**) B = 0.3 T and for two imposed potential E = −1.6 V/SCE and E = −1.8 V/SCE.

Concerning the electrochemical reactions produced on the substrate, as previously described in our first paper [26], this mechanism was similar to the zinc electroplating in an acidic sulfate medium containing Pb (II) ions [27].

These zinc–manganese alloy coatings elaborated in Figure 1 were characterized by XRD analyses. The diffractograms were normalized by taking the more intense peak which is the (101) orientation (Figure 2) as reference. The X-ray patterns allowed us to identify two phases η (JCPDS = 96-901-3473) and $MnZn_3$ (JCPDS = 96-153-8161) which have been already observed by Allam et al. [26] with deposition time equal to 10 min.

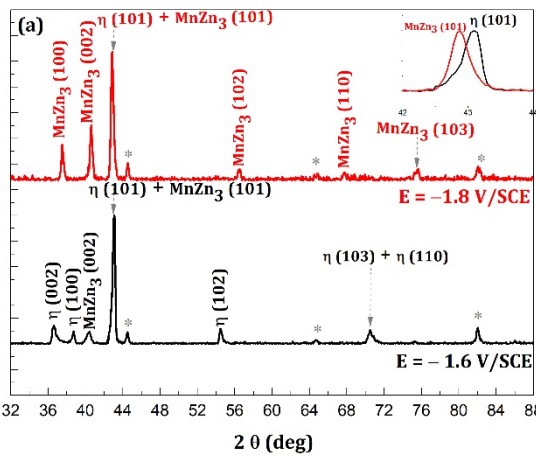

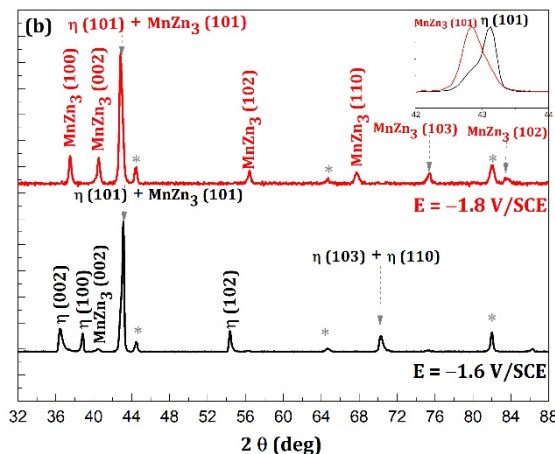

**Figure 2.** X-ray diffraction patterns of Zn–Mn alloy coatings electrodeposited as a function of magnetic field amplitudes: (**a**) B = 0 T, (**b**) B = 0.3 T and for two imposed potential E = −1.6 V/SCE and E = −1.8 V/SCE; asterisk represents substrate peaks. Phases η (JCPDS = 96-901-3473) and MnZn₃ (JCPDS = 96-153-8161).

For a constant magnetic field amplitude, the imposed potential modifies the structural composition of Zn–Mn alloy coatings. For E = −1.6 V/SCE (Figure 2a,b), the DRX diffractograms display a textured η phase with high intensity peaks and a textured MnZn₃ phase with very low intensity peaks. For high cathodic potential (E = −1.8 V/SCE) (Figure 3a) the coatings were also biphasic but, in this case, the MnZn₃ peaks were more intense than those of the η phase.

For a constant applied potential, the DRX diffractograms (Figure 2 and magnification) showed that the magnetic field superimposition lead to a decrease of the MnZn₃ peak intensities.

The chemical compositions of the Zn–Mn alloy coatings were analyzed by EDS as a function of imposed potentials and magnetic field amplitudes (Table 1).

**Table 1.** Chemical composition of Zn–Mn alloy coatings (at (%)) obtained by EDS for two imposed potentials E = −1.6 V/SCE and E = −1.8 V/SCE and for two magnetic field amplitudes B = 0 T and B = 0.3 T. The standard error is around 1 at % for all values.

| E (V/SCE) | B = 0 T | B = 0.3 T |
|:---:|:---:|:---:|
| −1.6 | Zn = 95%<br>Mn = 5% | Zn = 94%<br>Mn = 6% |
| −1.8 | Zn = 70%<br>Mn = 30% | Zn = 81%<br>Mn = 19% |

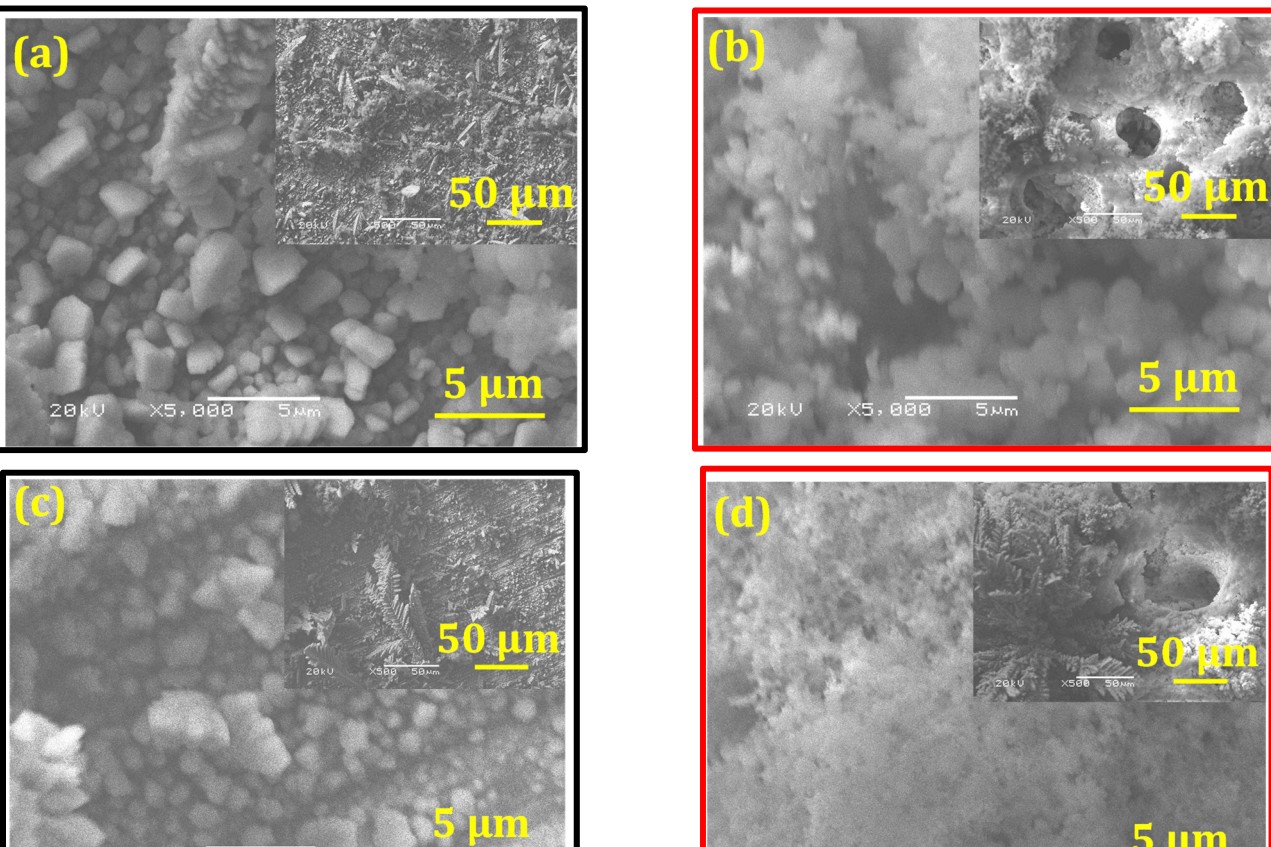

**Figure 3.** SEM micrographs of Zn–Mn alloy coatings electrodeposited as a function of magnetic field amplitudes: B = 0 T (**a**) E = −1.6 V/SCE and (**b**) E = −1.8 V/SCE, B = 0.3 T (**c**) E = −1.6 V/SCE, and (**d**) E = −1.8 V/SCE.

Table 1 shows that whatever the magnetic field amplitudes, the imposed potential strongly affected the zinc and manganese contents in the coating. At B = 0 T and B = 0.3 T, for cathodic potential E = −1.6 V/SCE, the percentage of manganese was close to 6 at%. However, increasing cathodic potential at E = −1.8 V/SCE increased this percentage to 30 at% without superimposed magnetic field and a much smaller increase was observed with superimposed magnetic field, the manganese content (19 at%) being lower than that obtained on the deposits made without a magnetic field.

This highest atomic Mn content (30 at%), correlated with the previous EDS and XRD measurements, seems to show that the $MnZn_3$ phase obtained in our deposits was not rigorously a $MnZn_3$ compound but a mixture of compounds of $Mn_{1+x} Zn_3$ formulas.

SEM micrographs of Zn–Mn alloy coatings are presented in Figure 3. It is shown that in the absence of magnetic field and for E = −1.6 V/SCE (Figure 3a) the coating highlighted a grains morphology. However, the homogeneity of the surface was lost for E = −1.8 V/SCE and the presence of pores was observed. Similarly, in the presence of magnetic field (Figure 3c,d), it was observed that the morphology variations for the two potential values followed substantially the same behavior when compared to those obtained in the absence of a magnetic field (Figure 3a,b), the similar current evolution leading to the same effects on morphologies, consequently, the actual surface was not changed [24].

### 3.2. Corrosion Behavior

#### 3.2.1. Long Immersion Time

Figure 4 shows the open-circuit potential (OCP) evolution for the ZnMn coatings and the substrate immersed in aerated NaCl 3.5% as a function of immersion time. All

curves (Figure 4a,b) showed a similar trend to that observed by Bučko et al. and Ganesan et al. [12,28] with three distinct zones. Initially, the corrosion potential was noticeably constant; it was characteristic of the Zn–Mn alloy chemical composition (see Table 1). This first zone was followed by a rapid rise in potential leading to a plateau. This rapid rise was related to the fact that the corroded alloy deposited present open pores, which allowed a contact between the substrate and the electrolyte. Finally, the potential stabilized at a constant value around −0.6 V/SCE, characteristic of corroded steel coated with corrosion products [12]. A contrary result was observed in the work of Tomić et al. [10] where this last step had a potential equal to that of steel; the authors explain that this was due to the loss of the layer of corrosion products formed during immersion.

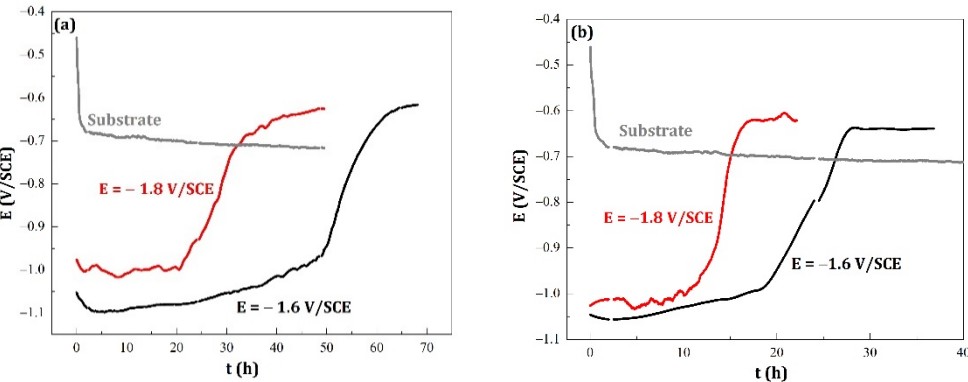

**Figure 4.** Evolution of corrosion potential in 3.5% NaCl solution versus time for carbon steel substrate and for Zn–Mn alloy coatings electrodeposited at (**a**) B = 0 T, (**b**) B = 0.3 T and at two imposed potential E = −1.6 V/SCE and E = −1.8 V/SCE.

Based on these results, the following points can be highlighted:

- For the same conditions of imposed potential during the electrodeposition, the super-imposition of the magnetic field equal to 0.3 T caused a decrease of $t_{lim}$ ($t_{lim}$ is the time for which the corrosion potential ($E_{corr}$) reaches its limit value).
- The results of the corrosion behavior of deposits developed without a magnetic field are in good agreement with the literature. Indeed, Ganesan et al. [28] and Bučko et al. [12] reported that alloy coatings elaborated at E = −1.6 V/SCE (low current density) showed better corrosion resistance than those at E = −1.8 V/SCE (high current density). This can also be related to the morphology of the deposits (Figure 3) and the presence of open pores, accelerated by the coating dissolution. These characteristics are related to lower corrosion resistance [12–14,16].
- Regarding the corrosion potential at immersion, there are two slightly different values, but differences are reproducible. For the coatings rich in zinc, and having the η phase in the majority, the $E_{corr}$ value is of the order of −1.05 V/SCE, while for the coatings rich with $MnZn_3$ phase (therefore, richer in manganese), the $E_{corr}$ value is higher with an average value equal to −1.00 V/SCE. This result is similar to those found in the literature [10,29].

For a more detailed numerical analysis of the curves obtained in Figure 4, a study of the open-circuit potential (OCP) curves was carried out. Due to the shape of the curves, the relationship between $E_{corr}$ and the immersion time can be written according to Equation (1).

$$E_{corr} = E_1 + E_2 \mathrm{erf}\left(\frac{t - t_1}{t_2}\right) \tag{1}$$

where: erf (x) is error function of x. $E_1$ and $E_2$ are related to the initial and final corrosion potentials, respectively. $E_{corr}$ (t = 0) ≈ $E_1$ − $E_2$ and $E_{corr}$ (t = ∞) ≈ $E_1$ + $E_2$. $E_1$ corresponds to the corrosion potential at $t_1$ and $t_1$ is the time for which the derivative $dE_{corr}/dt$ is maximum.

The duration $t_2$ is characteristic of the rate of the corrosion potential evolution between the initial and final potential values.

Note that the MSE value (mean square error), which is the average of the deviations squared, is expressed in Equation (2).

$$\text{MSE} = \frac{1}{n} \sum_{i=1}^{n} \left( \text{Ecorr}_{\text{experimental}} - \text{Ecorr}_{\text{calculated}} \right)^2 \tag{2}$$

with n the total number of values.

Figure 5 presents a modeling example using Equation (1).

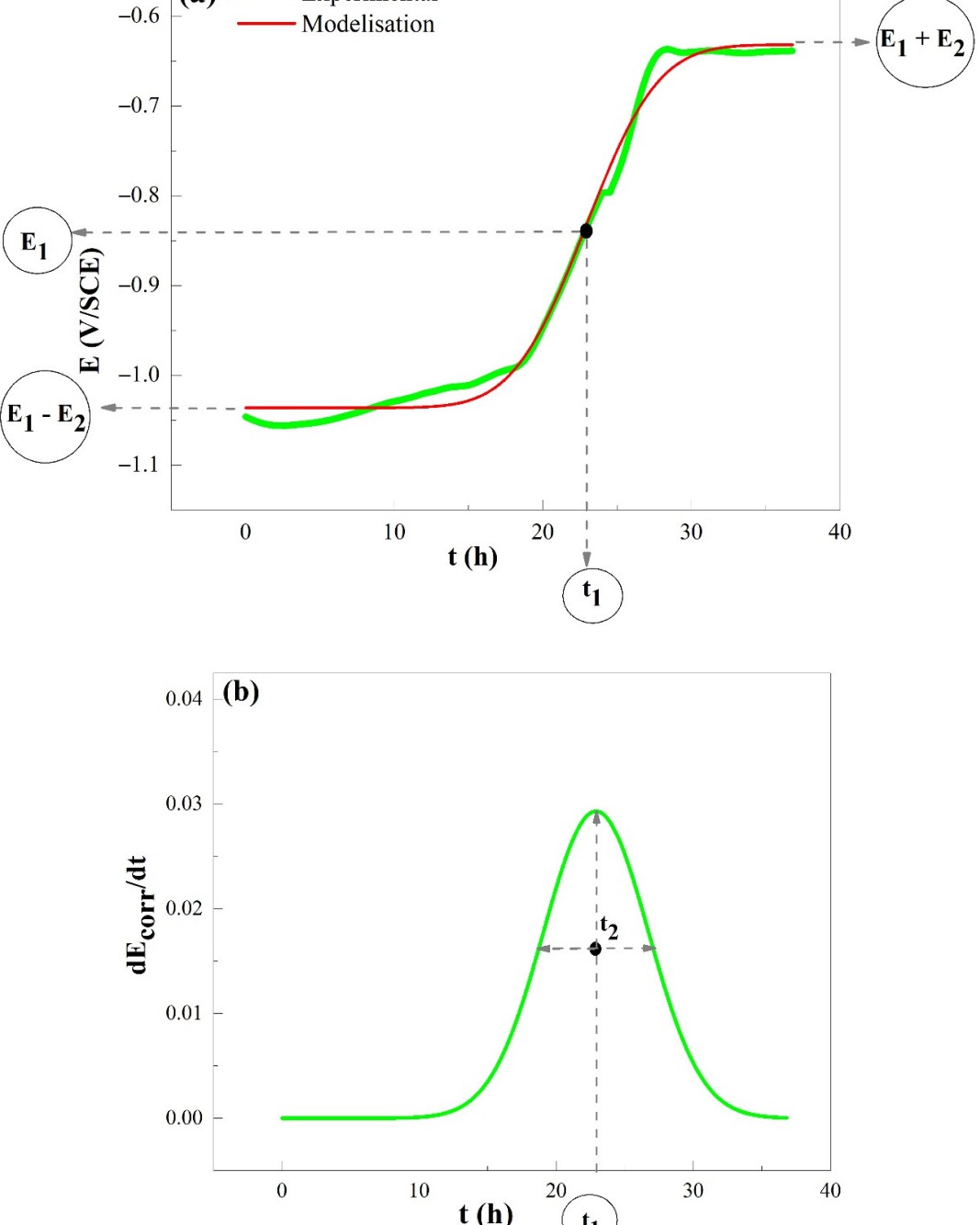

**Figure 5.** Example of modeled curve of open-circuit potential (**a**): simulation and (**b**) derivative (elaboration parameters: t = 30 min, E = −1.6 V/SCE, and B = 0.3 T).

It can be observed from Table 2 that:

- Whatever the magnetic field amplitude, $t_1$ is greatly decreased when the cathodic potential increased.
- Whatever the imposed potential, the superimposition of the magnetic field causes a 50% decrease of $t_1$ while the quantities of electric charge are either constant (for E = −1.8 V/SCE) or reduced by only about 20% (for E = −1.6 V/SCE) (see Figure 1);
- The rate of variation of the corrosion potential in the increasing section is almost constant for deposits elaborated without superimposition of magnetic field; on the other hand, it is greatly increased when a magnetic field is superimposed for B = 0.3 T during deposition and when the cathodic potential is equal to −1.8 V/SCE; and
- The term value $(E_1 - E_2)$ shows the same variations as those observed in the experimental curves (Figure 4), namely, a lower value for deposits rich in zinc (E = −1.6 V/SCE).

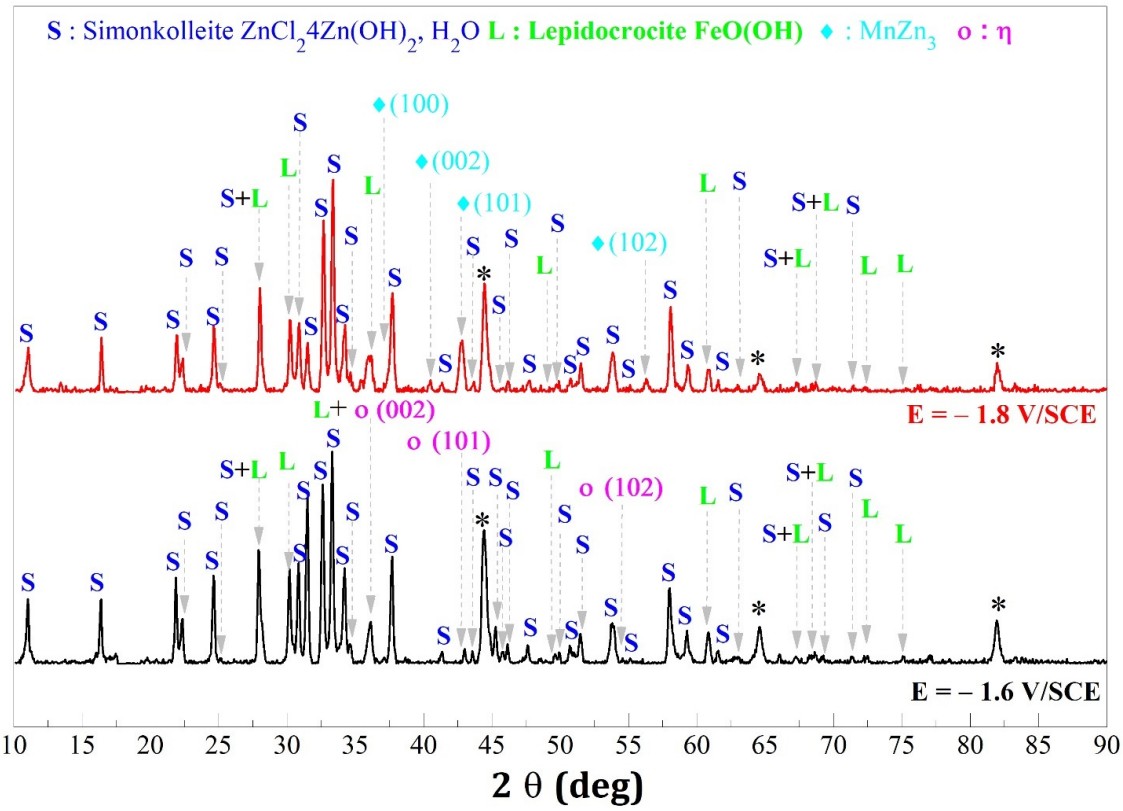

**Figure 6.** X-ray diffraction patterns of the deposits elaborated with and without the magnetic field and for two imposed potentials (**a**) E = −1.6 V/SCE and (**b**) E = −1.8 V/SCE after immersion for 48 h in 3.5% NaCl. Simonkolleite (JCPDS 00-007-0155) and lepidocrocite (JCPDS 01-074-1877).

**Table 2.** Modeling values of Zn–Mn alloy coatings behavior in 3.5% NaCl (see Figure 4).

| Elaboration Parameters | | Modeling Parameters | | | | | |
|---|---|---|---|---|---|---|---|
| B (T) | E (V/SCE) | $E_1$ (V/SCE) | $E_2$ (V/SCE) | $t_1$ (h) | $t_2$ (h) | $\dfrac{dE_{corr}}{dt_{max}}$ | $10^5 \cdot$ MSE |
| 0 | −1.6 | −0.827 | 0.247 | 53.0 | 11.6 | 0.024 | 35 |
| | −1.8 | −0.822 | 0.181 | 28.3 | 6.9 | 0.030 | 7.4 |
| 0.3 | −1.6 | −0.834 | 0.202 | 22.9 | 5.4 | 0.042 | 17.9 |
| | −1.8 | −0.814 | 0.199 | 14.1 | 2.2 | 0.104 | 12 |

The surface samples of coatings were characterized by XRD when the corrosion potential reached its maximum limit. The obtained diffractograms are presented in Figure 6. It is shown that the same chemical compounds were obtained for the deposits elaborated

with or without the magnetic field, namely, simonkolleite $ZnCl_2 \cdot 4Zn(OH)_2 \cdot H_2O$ (JCPDS 00-007-0155) and lepidocrocite $FeO(OH)$ (JCPDS 01-074-1877) with still the presence of the initial deposit (majority phase η and majority phase $MnZn_3$ for a deposition potential respectively equal to $-1.6$ V/SCE and $-1.8$ V/SCE). The absence of manganese corrosion products was already observed by Müller et al. and Bučko et al. [12,29]. The authors explained that either these oxides were amorphous or the manganese salts were soluble in the corrosive solution. Indeed, Bučko et al. added that a passivation layer composed only of hydroxy zinc chloride increased the corrosion stability of Zn–Mn coatings [12].

### 3.2.2. Short Immersion Time

The Nyquist and Bode impedance plots for Zn–Mn alloy coatings electrodeposited at two cathodic potentials and two magnetic field amplitudes are shown in Figures 7 and 8.

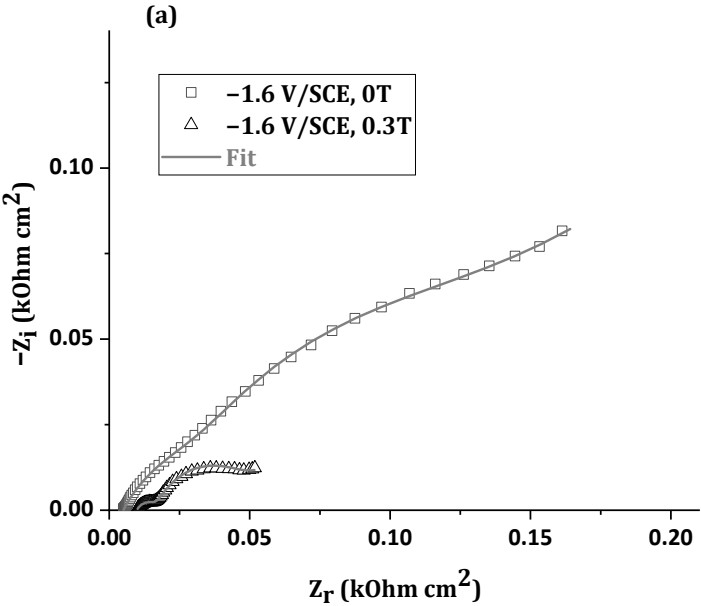

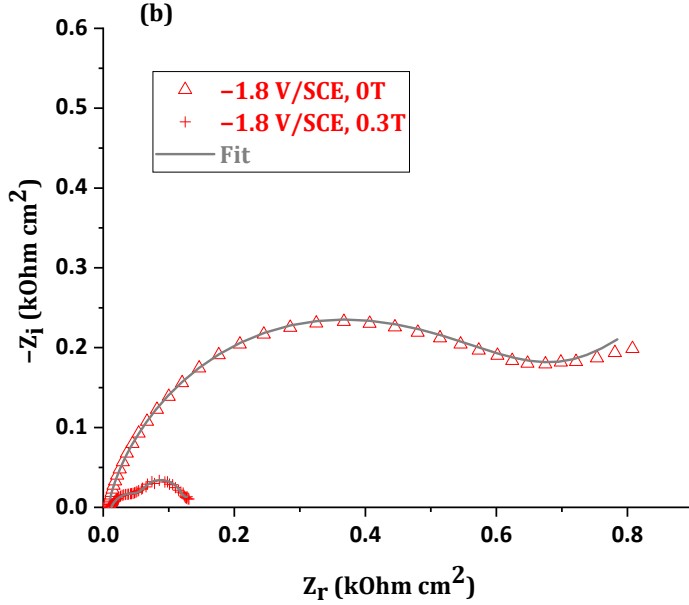

**Figure 7.** Nyquist plots for Zn–Mn alloy coatings electrodeposited with and without the magnetic field and for two imposed potentials (**a**) E = $-1.6$ V/SCE and (**b**) E = $-1.8$ V/SCE after 2 h of immersion in 3.5% NaCl solution.

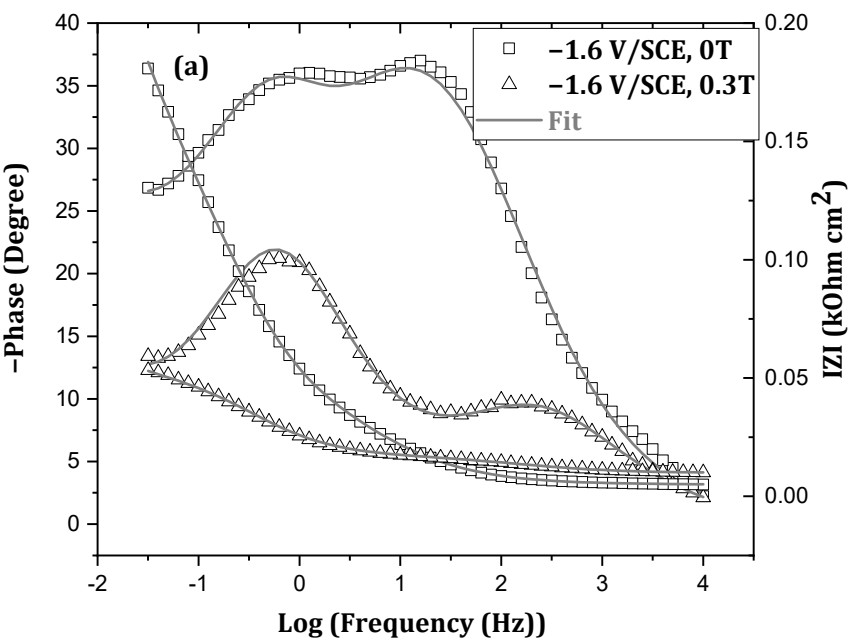

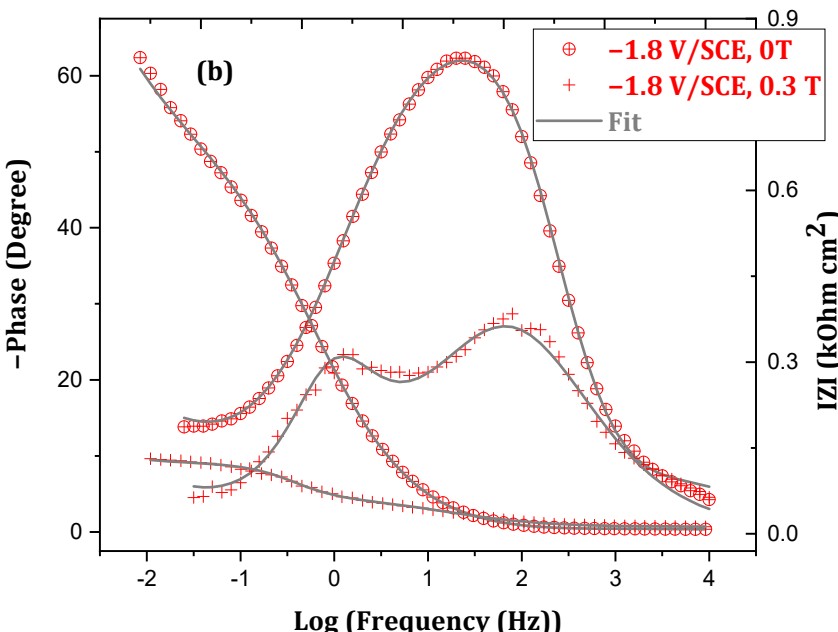

**Figure 8.** Bode plots for Zn–Mn alloy coatings electrodeposited for (**a**) B = 0 T, (**b**) B = 0.3 T and at two imposed potential E = −1.6 V/SCE and E = −1.8 V/SCE after 2 h of immersion in 3.5% NaCl solution.

Whatever the imposed potential, it is visible that the behavior of the deposited alloys is strongly impacted by the presence of the magnetic field during electrodeposition. These results can be analyzed by modeling a classical equivalent electrical circuit representative of corrosion phenomena [30] shown in Figure 9. This circuit can be used and the parameters obtained using ZSimpWin, as given in Table 3.

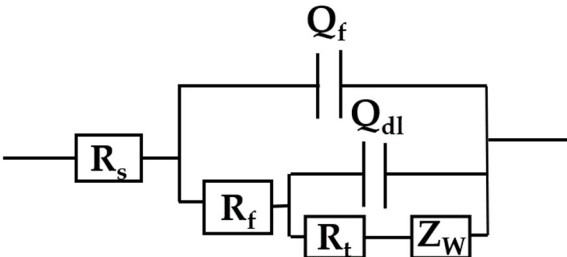

**Figure 9.** Equivalent electrical circuit used for electrochemical impedance modeling presented in Figure 8.

$R_s$ is the solution resistance between the reference electrode (SCE) and the working electrode (CE).

$R_f$ represents the resistance of the layer containing the corrosion products formed on the surface (according to simonkolleite in Figure 6).

$R_t$ is attributed to the charge transfer resistance of the corrosion mechanism.

$Q_f$ is the constant phase element (CPE) relative to the capacity of the corroded ZnMn alloy coating.

$Q_{dl}$ is the CPE element relating to the electrochemical double layer capacity.

W is the Warburg impedance characteristic of the oxygen diffusion phenomenon.

As the actual surfaces of the deposits are not identical (see roughness values in Table 4), the obtained values cannot be easily compared with each other. However, several findings can be drawn regarding the effect on the corrosion properties of deposits.

**Table 3.** Optimum fit parameters for Zn–Mn alloy deposited as a function of cathodic potential and magnetic field.

| E (V/SCE) | B (T) | $R_s$ ($\Omega\cdot cm^2$) | $Q_f$ $10^3\ Y_f$ ($Ss^{nf}\ cm^{-2}$) | $n_f$ | $R_f$ ($\Omega\cdot cm^2$) | $Q_{dl}$ $10^3\ Y_{dl}$ ($Ss^{ndl}\ cm^{-2}$) | $n_{dl}$ | $R_t$ ($\Omega\cdot cm^2$) | $10^2\ W$ ($\Omega\cdot s^{-0.5}\ cm^{-2}$) | $10^3\ \chi^2$ |
|---|---|---|---|---|---|---|---|---|---|---|
| −1.6 | 0 | 5.0 | 3.1 | 0.7 | 65.6 | 4.50 | 0.9 | 101 | 2.62 | 0.30 |
|  | 0.3 | 9.7 | 2.2 | 0.6 | 8.73 | 14.5 | 0.8 | 30.6 | 19.6 | 0.14 |
| −1.8 | 0 | 7.7 | 0.28 | 0.8 | 491 | 1.53 | 0.8 | 197 | 1.10 | 3.0 |
|  | 0.3 | 12 | 0.57 | 0.7 | 50.0 | 3.48 | 0.9 | 67.5 | 26.0 | 0.74 |

**Table 4.** Roughness values of the coatings elaborated during 30 min before and after 2 h immersion in 3.5% NaCl.

| Elaboration Parameters | E = −1.6 V/SCE | | E = −1.8 V/SCE | |
|---|---|---|---|---|
|  | **B = 0 T** | **B = 0.3 T** | **B = 0 T** | **B = 0.3 T** |
| Rms before immersion (µm) | 7.9 ± 0.3 | 8 ± 0.3 | 10.3 ± 0.1 | 16.1 ± 0.6 |
| Rms after 2 h immersion (µm) | 8.4 ± 0.3 | 19.3 ± 0.2 | 14.4 ± 0.2 | 20.8 ± 0.5 |

As it can be seen from Table 3, whatever the imposed potential (E = −1.6 V/SCE or E = −1.8 V/SCE), the magnetic field causes a decrease in the two parameters $R_f$ and $R_t$, which reflects a greater electrochemical reactivity correlated with the significant decrease in the corrosion resistance time of the considered deposits. Similarly, changes in $R_f$ and W values due to the magnetic field superimposed during electrodeposition show changes in the corrosion mechanism of the alloy.

Moreover, the roughness measurements (Table 4) show that the corrosion phenomenon systematically increases the roughness of the deposit with a maximum value obtained after the corrosion for the deposits elaborated under high cathodic potential and magnetic field equal to B = 0.3 T (Rms = 20.8 ± 0.5 µm). This roughness increase can be also a reason for the $R_f$ and $R_t$ decreasing due to the modification of the actual reactive surface.

Figures 10–13 show SEM micrographics of ZnMn coatings. For E = −1.6 V/SCE and whatever the magnetic field amplitude superimposed during the elaboration, there is no significant changes in morphology and chemical composition before and after corrosion immersion for two hours (Figures 10 and 12). On the other hand, for the samples elaborated for higher cathodic potential (E = −1.8 V/SCE), the corrosion phenomenon affects not only the morphology (Figures 11 and 13) but also the chemical composition with the atomic Mn ratio, which significantly decreases from 31 to 17 at% and from 30 to 10 at% for the deposits obtained at B = 0 T and B = 0.3 T, respectively.

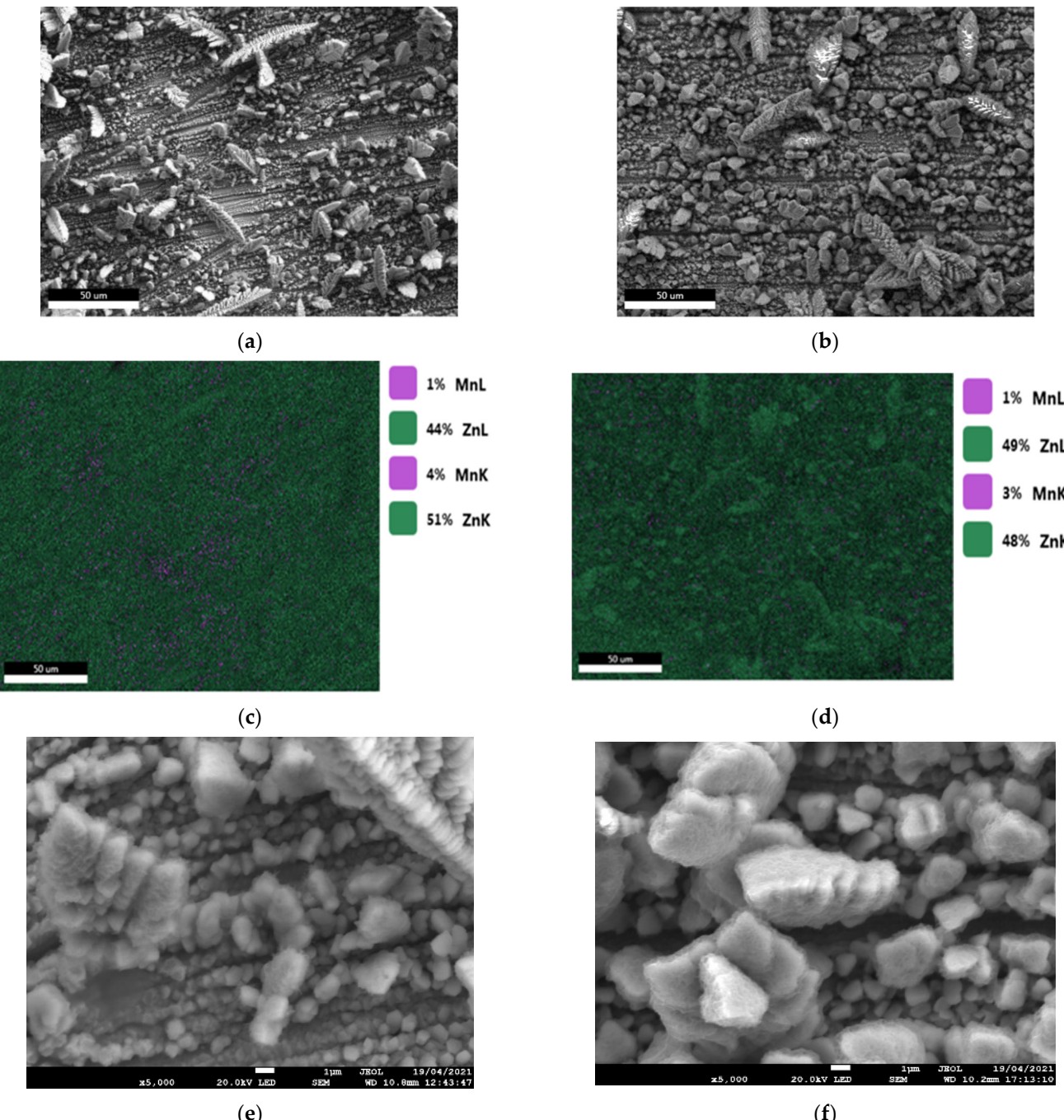

**Figure 10.** Morphologies of coatings elaborated for B = 0 T and for E = −1.6 V/SCE, before immersion ((**a**) image × 500, (**b**) mapping, and (**c**) image × 5000 and after immersion for 2 h in 35 g/L NaCl (**d**) image × 500, (**e**) mapping, and (**f**) image × 5000).

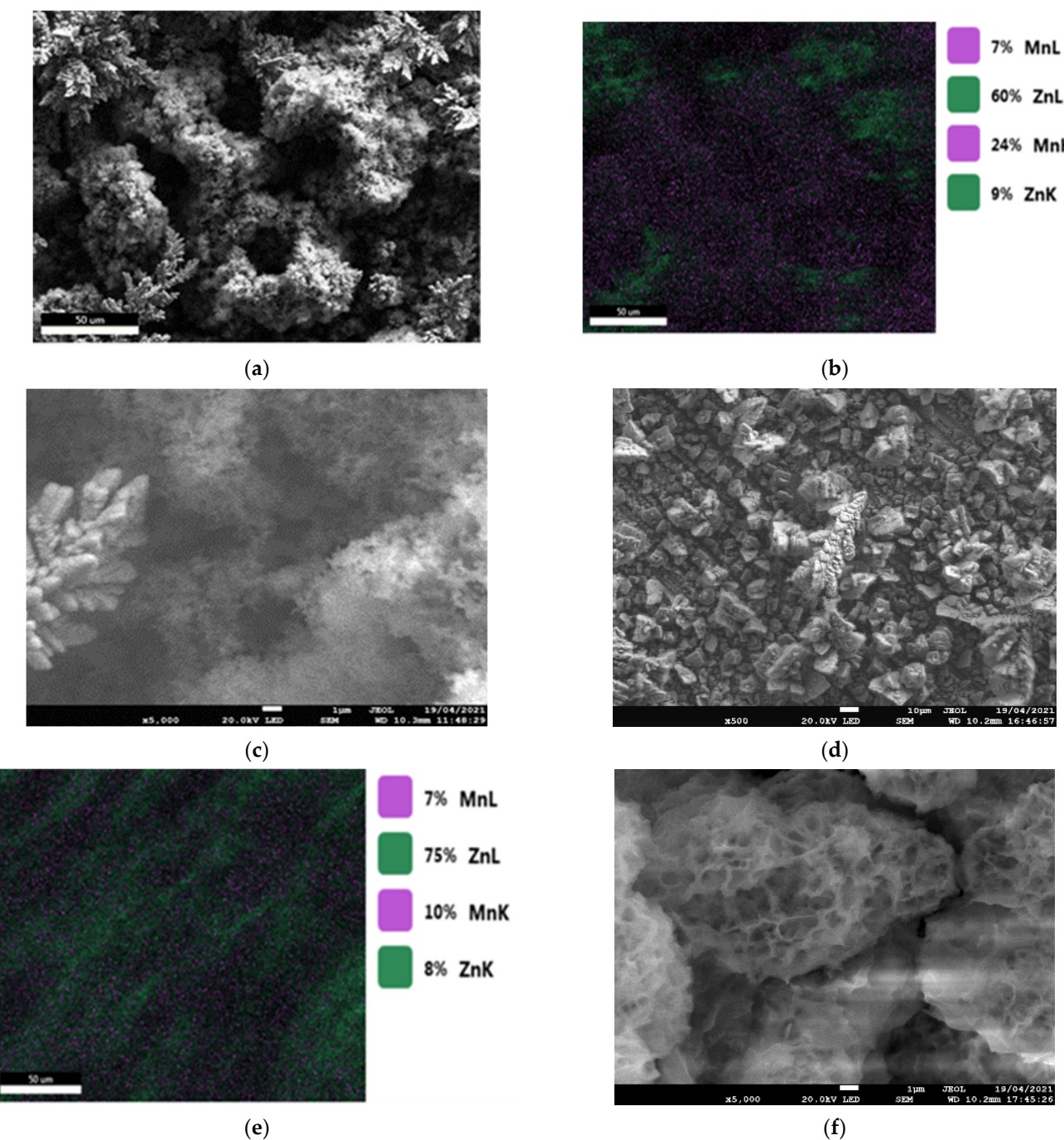

**Figure 11.** Morphologies of coatings elaborated for B = 0 T and for E = −1.8 V/SCE, before immersion ((**a**) image × 500, (**b**) mapping, and (**c**) image × 5000 and after immersion for 2 h in 35 g/L NaCl (**d**) image × 500, (**e**) mapping, and (**f**) image × 5000).

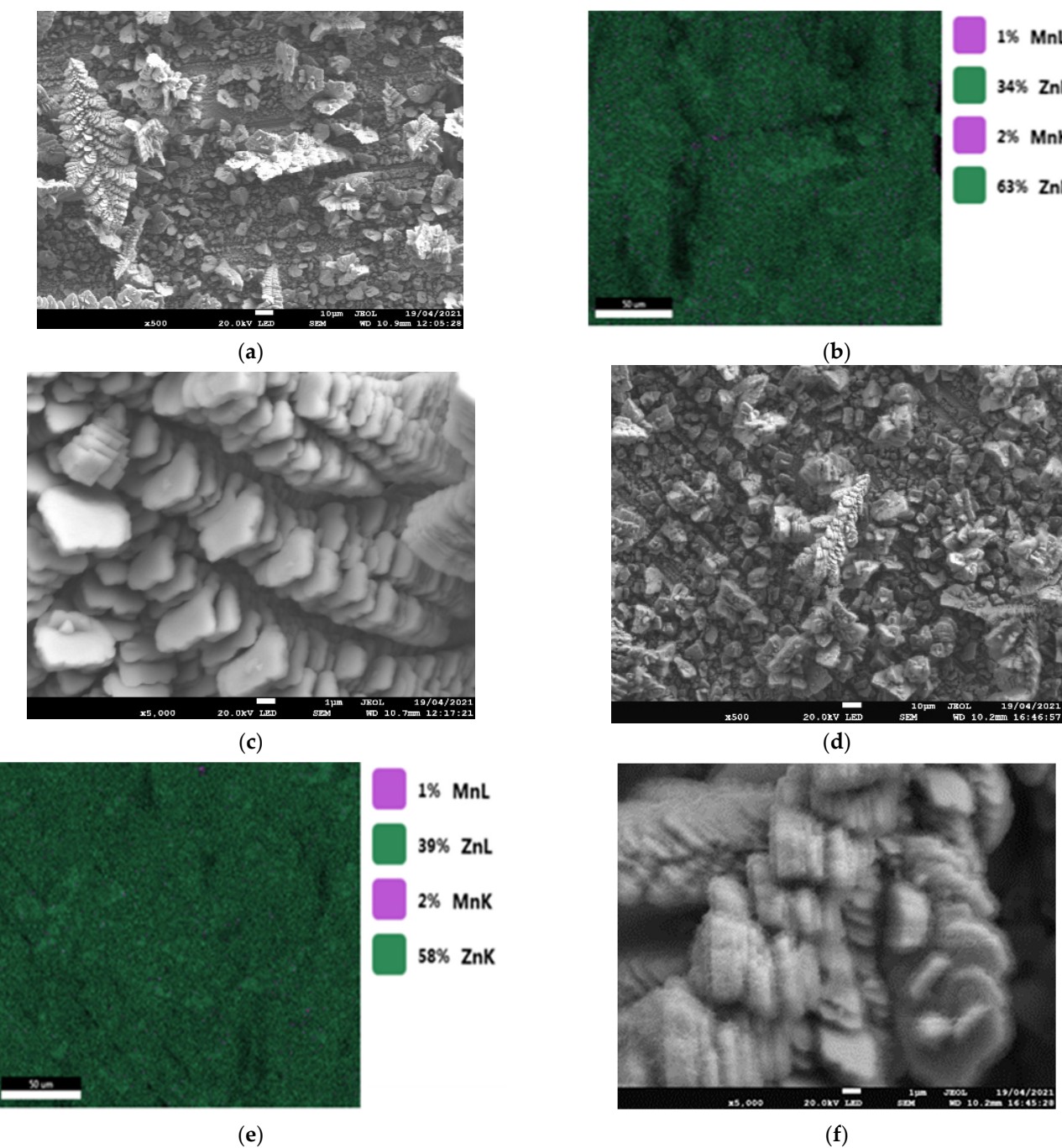

**Figure 12.** Morphologies of coatings elaborated for B = 0.3 T and for E = −1.6 V/SCE, before immersion ((**a**) image × 500, (**b**) mapping, and (**c**) image × 5000 and after immersion for 2 h in 35 g/L NaCl (**d**) image × 500, (**e**) mapping, and (**f**) image × 5000).

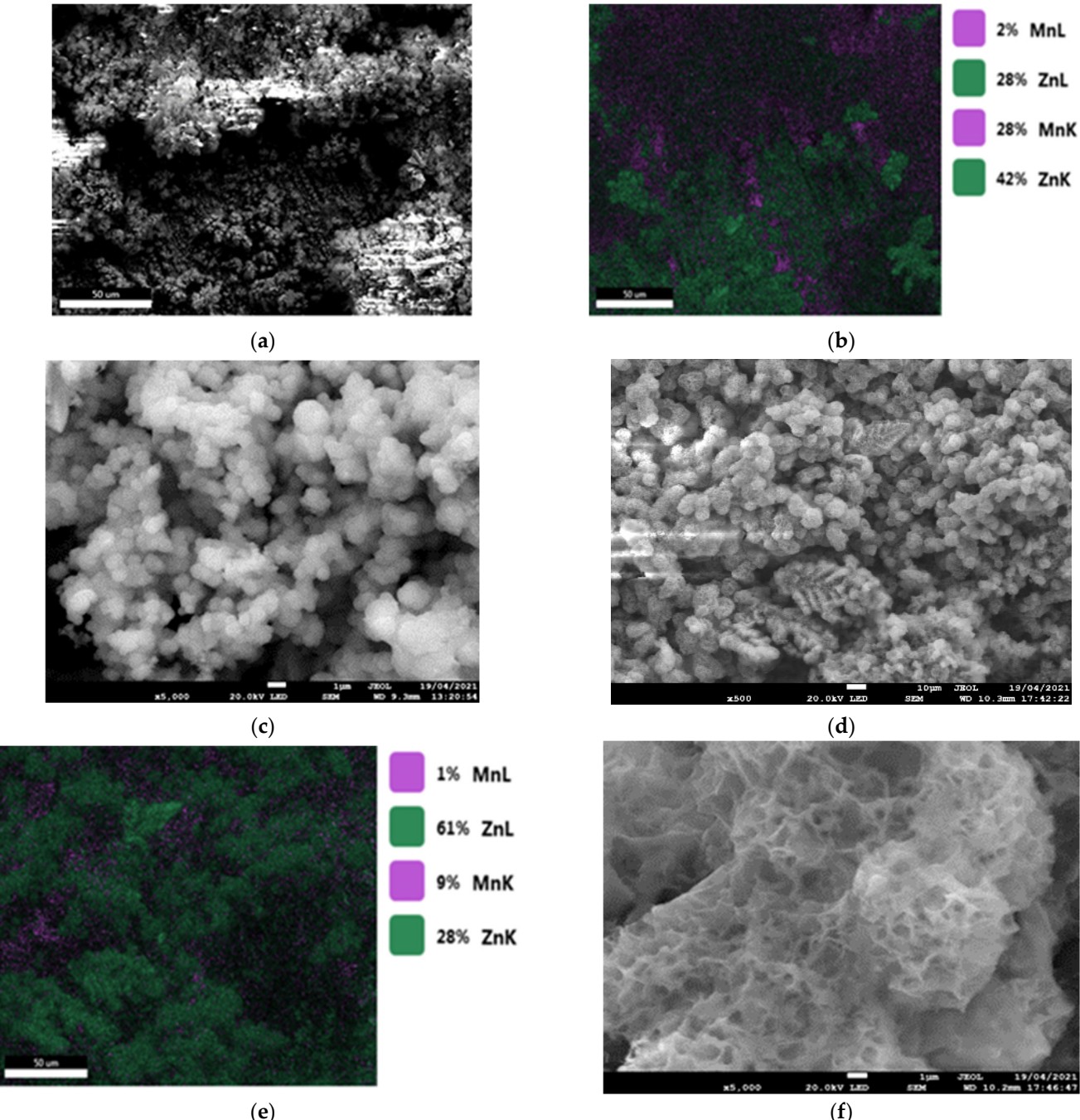

**Figure 13.** Morphologies of coatings elaborated for B = 0.3 T and for E = −1.8 V/SCE, before immersion ((**a**) image × 500, (**b**) mapping, and (**c**) image × 5000 and after immersion for 2 h in 35 g/L NaCl (**d**) image × 500, (**e**) mapping, and (**f**) image × 5000).

The X-ray diffraction analyses before and after corrosion show the structural evolution of the deposits due to the corrosion phenomenon (Figures 14 and 15). We can notice that after 2 h of immersion, a very weak signal related to the appearance of the corrosion product indexed "S" appears in the diffractograms. This product could be determined as simonkolleite (JCPDS 00-007-0155).

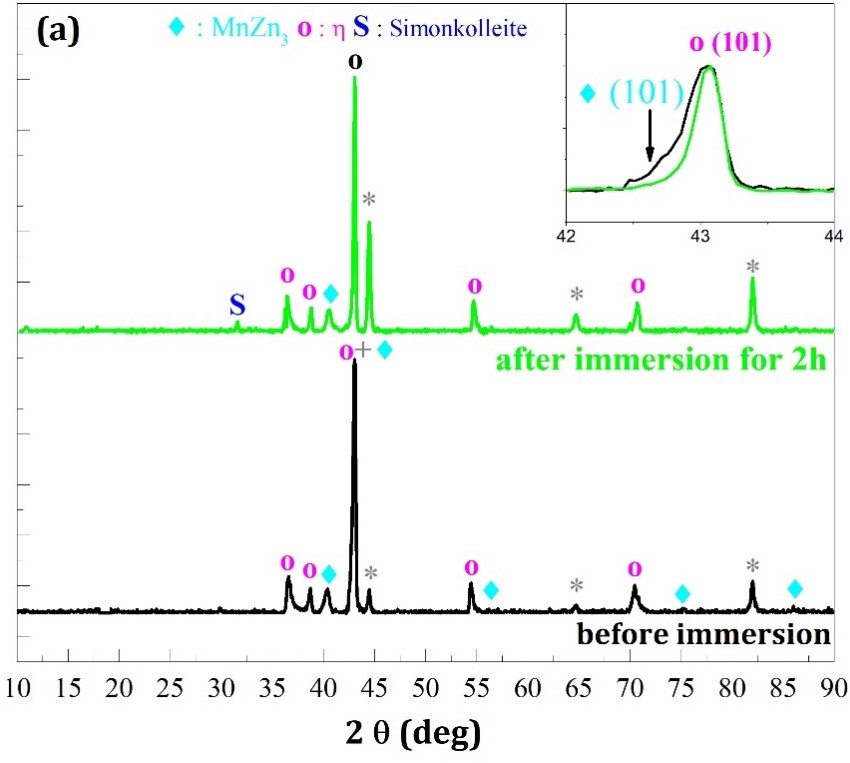

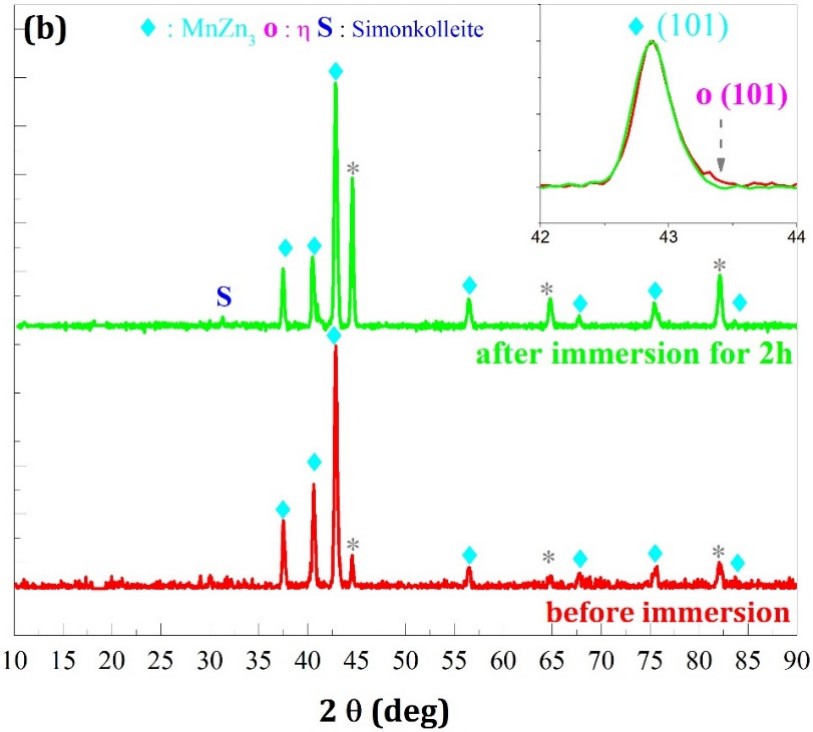

**Figure 14.** X-ray diffraction patterns of the deposits elaborated for B = 0 T and for two imposed potentials (**a**) E = −1.6 V/SCE and (**b**) E = −1.8 V/SCE before and after immersion for 2 h in 3.5% NaCl. Simonkolleite (JCPDS 00-007-0155), phases η (JCPDS = 96-901-3473), and MnZn$_3$ (JCPDS = 96-153-8161).

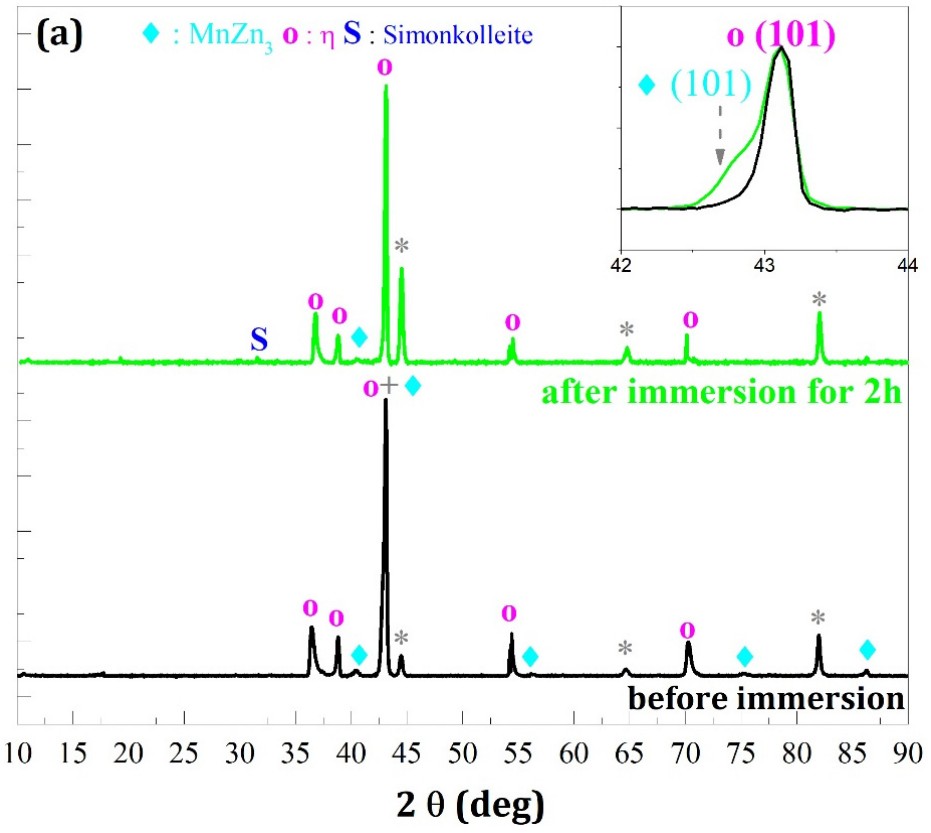

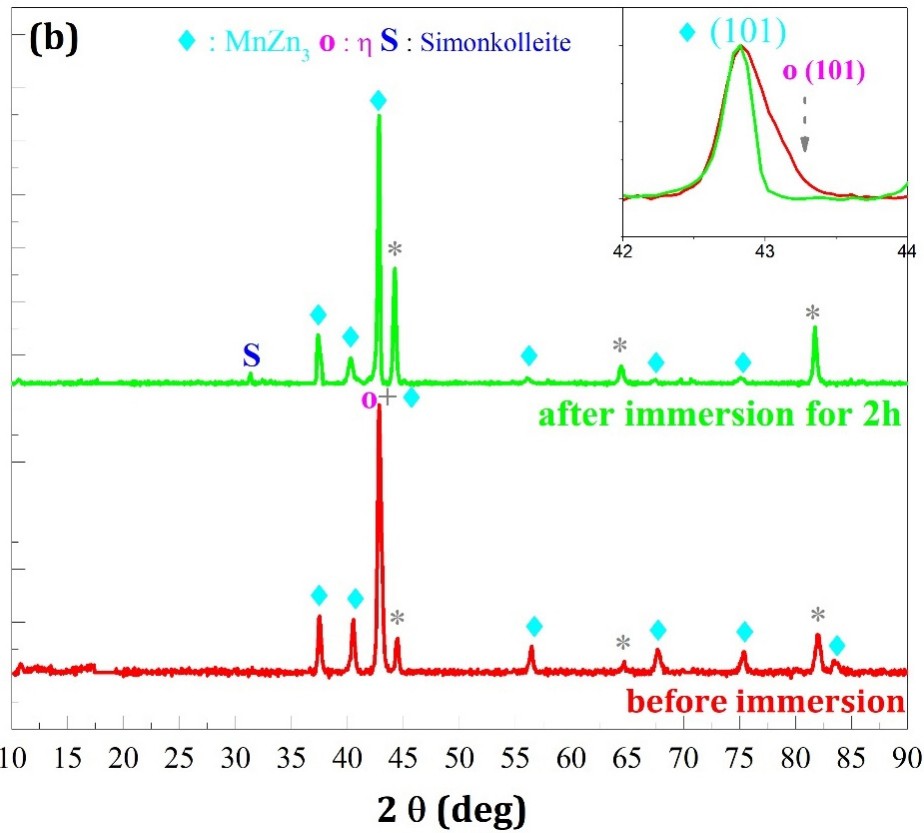

**Figure 15.** X-ray diffraction patterns of the deposits elaborated for B = 0.3 T and for two imposed potentials (**a**) E = −1.6 V/SCE and (**b**) E = −1.8 V/SCE before and after immersion for 2 h in 3.5% NaCl. Simonkolleite (JCPDS 00-007-0155), phases η (JCPDS = 96-901-3473), and $MnZn_3$ (JCPDS = 96-153-8161).

Based on these analyses, the following points can be deducted:

- In the case where the η phase is majority (cathodic potential deposition equal to −1.6 V/SCE) (Figures 14a and 15a) the relative intensities of the η phase peaks are not modified with respect to each other, whereas for the minority phase MnZn$_3$ we observe a disappearance of the characteristic line (101) and a decrease of the line (002) which are more important for the deposition carried out under magnetic field (Figure 15a).

- For the deposits carried out at a potential of −1.8 V/SCE (Figures 14b and 15b), the same observation on the selectivity of the corrosive attack can be made. Indeed, for the deposit carried out without magnetic field (Figure 14b), the relative intensity of the line (002) of the MnZn$_3$ phase decreases strongly whereas for all the other lines the intensities are appreciably identical before and after corrosion. Finally, for the deposit elaborated under magnetic field (B = 0.3 T), we can observe the disappearance of the (101) line of the η phase leading to a monophasic deposit with a more textured MnZn$_3$ phase after corrosion since all the relative intensities of these phases, except (100) and (101), have strongly decreased or even have disappeared. These structural modifications are to be put in parallel with the previous observations obtained by EDS which show differences in the rate of disappearance of the element manganese according to the electrodeposition process.

## 4. Conclusions

This study of the properties of magnetoelectrodeposited Zn–Mn alloy coatings and their behavior in corrosion allows us to conclude the following points:

1.  For deposits obtained for substantially equal current densities, mainly presenting majority the η phase (deposits made at E = −1.60 V/SCE) with a percentage of zinc greater than 90% and identical XRD diffractograms, the behavior against corrosion depends on the electrodeposition process (superimposition, or not, of the magnetic field);

2.  For identical chemical compositions of coatings, the superimposition of a magnetic field greatly decreases their corrosion resistance; and

3.  The same observation can be noted for Zn–Mn coatings having a high proportion of the MnZn$_3$ phase with zinc proportions less than 90% where, in addition to the corrosion-resistance decrease, a significant increase in the speed of the corrosion potential transition stage has been observed.

These findings show that if the rich deposits in η phase have a much better resistance to corrosion than those rich in MnZn$_3$ phase, for the deposits having the same crystallographic structures, the corrosion behavior can be strongly modified by the magnetohydrodynamic (MHD) convection during a magnetoelectrodeposition process. These effects on the corrosion-resistance coatings cannot be easy correlated with the alloy chemical compositions, but with the nature of these alloys. This is highlighted by the comparison either the electrodeposition process at two different potentials and the same magnetic field amplitude (B = 0 T or B = 0.3 T) or by the comparison of deposits obtained at the same potential equal to −1.80 V/SCE for the two magnetic field values.

**Author Contributions:** L.A. performed the experiments, designed the experiments, analyzed and interpreted the data, and wrote the paper. F.S.L. contributed reagents and materials, designed the experiments, analyzed and interpreted the data, and corrected the paper. J.-P.C. contributed reagents and materials, designed the experiments, analyzed and interpreted the data, and corrected the paper. All authors have read and agreed to the published version of the manuscript.

**Funding:** This research received no external funding.

**Institutional Review Board Statement:** Not applicable.

**Informed Consent Statement:** Not applicable.

**Data Availability Statement:** Not applicable.

**Acknowledgments:** The authors warmly thank the "Programme Franco–Algérien "PROFAS B+ 2018" for the financial support through the grant awarded to one of the co-authors (grant number 931461K-958020D).

**Conflicts of Interest:** The authors declare no conflict of interest.

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
