# Peer review of "Corrosion Behavior of ZnMn Coatings Magnetoelectrodeposited"

_magnetochemistry, doi:10.3390/magnetochemistry8070069_

Round 1

Reviewer 1 Report

no comments

Author Response

No comments were provided from reviewer 1. 

Reviewer 2 Report

The effect of  magnetic field on the electrodepostion behavior is interesting, however, this manuscript has many unclear part. Here are the details. 

  1. The potential for deposition is -1.6 and -1.8, please point out the electrochemical reaction on the substrate, and then discuss the effect of magnetic field on those reaction. By the way, is there any bubbles during the electrodeposition process. 
  2. It doesn’t make sense from Eq. 1 to Fig. 5 and Table 2. The modeling for corrosion potential fitting should be paid more attention, please comment on it. 
  3. The magnetic field decreases the Rf and Rt, please give more explanation to this. The Unit in Table 3 is wrong. 
  4. Two many similar figures, please summarize results more effectively.

Reviewer 3 Report

In this paper, the authors studied the corrosion behavior of Zn-Mn alloy coatings obtained with and without magnetic field involved in. There are some issues needed to be addressed.

1) The format issues, such as in Figure 2(a), the bottom part is cutted. L345-L350, the font format is strange. 

2) The value in Table 1 is in weight percentage or volume percentage? Please provide the EDS plot so that we can have a clear obaservation at the data.

3) Figure 9 is unclear, please redraw it and expliain each element clearly.

4) How did you distinguish the L and K constitute in Figure 10 and the following? 

Reviewer 4 Report

Allam et al studied corrosion behavior of magnetoelectrodeposited ZnMn coatings. The work presents interesting results and can be considered for acceptance after minor revisions.

1. It is suggested that authors firstly state the problem to be solved or the purpose of the paper, and then describe the experimental results.

2. It is recommended that the authors add JCPDS card data into the XRD results.

3. Authors are suggested to consider recombining Figure 10-13 for a more significant comparison of data.

4. The title and numbers on the abscissa of FIG. 1 are not displayed completely, please check.

5. Some references are too old. Cite the related works in last two years. The following work is for reference:

[1] TRANSACTIONS OF NONFERROUS METALS SOCIETY OF CHINA 2021, 31 (6), 1842-1852. doi: 10.1016/S1003-6326(21)65621-2.

[2] MATERIALS SCIENCE AND TECHNOLOGY 2021, 37 (14), 1187-1198. doi: 10.1080/02670836.2021.1987696.

Round 2

Reviewer 2 Report

It can be accepted for publication.